

# Comparison of lower-leg muscle activation and establishment of muscle activation patterns during single-leg stance under various instability conditions in healthy active subjects: a cross-sectional study

Mariana Sánchez-Barbadora[1,2], Vicente Alepuz-Moner[2],
Noemi Moreno-Segura[2] and Rodrigo Martín-San Agustín[2]

[1] Faculty of Health Sciences, Universidad Europea, Valencia, Spain
[2] Physiotherapy Department, Universidad de Valencia, Valencia, Spain

## ABSTRACT

**Background:** Balance training exercises are essential for treating and preventing a variety of ankle conditions such as sprains, chronic ankle instability, and muscular weakness. This muscle activation could often be increased using unstable devices. The Blackboard is a new configurable selective instability device, that has the potential to focus directly on desired muscles without overloading others through wooden half cylinders slats joined to a board by a tape in different positions. Depending on the position where they are situated, different stability training could be performed. However, its specific muscle activation patterns remain unknown.

**Methods:** Surface electromyography was used to measure the activation level of six lower-leg muscles (*i.e.*, soleus, gastrocnemius medialis, gastrocnemius lateralis, tibialis anterior, peroneus longus, and peroneus brevis) in a single-leg stance on the floor and on seven Blackboard configurations. Thirty healthy active subjects participated in the study.

**Results:** Multiple differences in muscle activation were observed between conditions and among muscles. Notably, the tibialis anterior and the peroneus showed the greatest differences between conditions and the highest activation levels on the Blackboard. Additionally, forefoot supination and rearfoot eversion configurations induced selective activation of the tibialis anterior and peroneus longus, respectively, highlighting their usefulness for isolating specific muscle contractions.

Corresponding author
Noemi Moreno-Segura,
noemimorenosegura@gmail.com

## INTRODUCTION

Balance training exercises are essential for treating and preventing a variety of ankle conditions such as sprains, chronic ankle instability (CAI), and muscular weakness (*Bleakley et al., 2019*). These proprioceptive exercises are typically used either as isolated interventions or in combination with other therapeutic techniques (*e.g.*, manual therapy or electrostimulation) (*Czajka et al., 2014*), and can be performed under various conditions;

with eyes open or closed, while other tasks are performed (*e.g.*, passing a ball) (*Romero-Franco et al., 2014*), or even on different unstable devices (*Peña García-Orea et al., 2016*). Proprioception is an essential component of motor control and is considered an important characteristic for dynamic joint stability (*Riemann & Lephart, 2002*).

Unstable devices are commonly used in single-leg stance or in bipedal stance to improve stability, and have been proven effective in increasing lower-leg muscle activation during a rehabilitation program (*Braun Ferreira et al., 2011*; *Behm et al., 2015*; *Peña García-Orea et al., 2016*; *Wright, Linens & Cain, 2017*). This is due to an increasement in neuromuscular activation, a higher sensitivity of proprioceptors, higher motor control in cortex and a higher integration of the vestibular and visual system (*Büchel et al., 2021*; *Glass & Wisneski, 2023*). Traditionally, the most common devices used in this regard have been the BOSU® Balance Trainer (Hedstrom Fitness®, Ashland, OH, United States) and the Wobble Board (RPM Power, Thurles, Co. Tipperary, Ireland), which produce an undiscriminated global instability, in the multidirectional plane (*Stanek, Meyer & Lynall, 2013*; *Cuğ, Duncan & Wikstrom, 2016*; *de Silva et al., 2016*; *Strøm et al., 2016*). That means that the direction and intensity of the instability cannot be selected and adjusted by the user. Others, such as the PowerBalance Board (RPM Power, Thurles, Co. Tipperary, Ireland) or the half-foam roller, are designed to target a single plane of movement which can be frontal or sagittal depending on their placement (*Stanek, Meyer & Lynall, 2013*; *Peña García-Orea et al., 2016*; *Sánchez-Barbadora et al., 2022*). However, these devices are still global in terms of the targeted muscles, as the user is not able to configure them to focus the instability on the desired structure.

Recently, specialized instability devices have been developed to isolate and target specific areas, allowing the patient to focus on the desired movement. These devices aim to train specific proprioception without inducing fatigue, effusion, or pain, as these factors may negatively affect proprioception and motor learning (*Röijezon, Clark & Treleaven, 2015*). An example of this is the Mini Stability Trainer (Artzt®, Dornburg, Germany), which has been shown to specifically target certain muscles, producing significantly higher activation of the peroneus longus (PL) compared to other muscles or in comparison to other surfaces or conditions analyzed (*Alfuth & Gomoll, 2018*). Another similar device is the Blackboard (BB), which has been demonstrated to activate the PL as much as the rest of traditional global devices under study (*i.e.*, BOSU®, Wobble board, and PowerBalance Board) in healthy subjects (*Sánchez-Barbadora et al., 2022*). Both the Mini Stability Trainer and the BB apparently focus directly on the desired muscle without overloading others, thus enhancing the rehabilitation process by promoting targeted strengthening and balance improvement, which is crucial for effective recovery (*Sánchez-Barbadora et al., 2022*). In addition, recent studies have demonstrated the effectiveness of this device in improving dynamic balance, showing results comparable to those of the most commonly used device, the BOSU (*Sánchez-Barbadora et al., 2024*, *2025*).

Training the forefoot and rearfoot separately could allow for optimized muscle activation, improved postural control, and injury prevention by addressing biomechanical imbalances. Previous studies have shown that rearfoot eversion training increases peroneal activation, contributing to lateral ankle stability and reducing the incidence of sprains

(*Holmes & Delahunt, 2009*). Conversely, forefoot pronation training strengthens intrinsic foot muscles and the peroneus longus, improving arch function and propulsion efficiency (*Fraser, Feger & Hertel, 2016*). Implementing platforms designed to independently train these segments may be an effective strategy for injury rehabilitation and increasing performance in activities requiring precise balance and foot control. In this context, the stabilization of the ankle and foot involves several key muscles, including the aforementioned PL, as well as the tibialis anterior (TA) and the peroneus brevis (PB) (*Karagiannakis, Iatridou & Mandalidis, 2020*). TA plays a vital role in dorsiflexion of the ankle and can be related with conditions such as foot drop when weakened (*Stewart, 2008*; *Jeon et al., 2013*) and its function is essential for proper gait and balance (*Maharaj, Cresswell & Lichtwark, 2019*). Additionally, peroneus muscles are crucial for ankle stability due their role in pronation (*Yoshida & Suzuki, 2020*), and their weakness or dysfunction can contribute to CAI and increase the risk of lateral sprains (*Thompson et al., 2018*). Additionally, these muscles have a relationship with other foot conditions such as the hallux valgus (*Perera, Mason & Stephens, 2011*; *Ikuta et al., 2024*), further highlighting their importance in maintaining overall foot and ankle health.

In this sense, and considering that BB device could increase the activation of these key muscles, it appears that could be an interesting and innovative device in the field of balance training. However, the muscular activation it generates in its different configurations is unknown, as are the patterns of muscular activation that take place during its use.

We hypothesized that: (1) inducing a specific instability in the foot using the BB will lead to a specific foot movement that will selectively activate specific muscles, such as the tibialis anterior in forefoot instability configuration, supination of the forefoot exercise and inversion of the rearfoot exercise, and the peroneus longus in forefoot instability configuration, pronation of the forefoot exercise and eversion of the rearfoot exercise; and (2) each configuration of the BB will produce its own pattern of muscle activation in terms of intensity and co-contraction according to the induced movement. In this sense, we expect that the activation of the agonist inhibits the activation of the antagonist (*i.e.*, when peroneus longus is highly activated, the tibialis anterior is not and *vice-versa*). Therefore, the present study aimed to achieve two main objectives; first, to compare the muscle activation of six lower-leg muscles during the single-leg stance exercise across different conditions both on the floor and in seven instability configurations of the BB, and second, to compare the muscle activation patterns of these six muscles produced under each condition.

## MATERIALS AND METHODS

### Participants

An *a priori* sample size estimation was performed using G * Power v.3.1.9.2 software. Based on *Alfuth & Gomoll (2018)*, who observed large muscle activation effect sizes, an expected effect size of F = 0.4 with α = 0.05 indicated a required sample size of 19 for >80% power. To increase measure power, the sample size was raised to 30 participants (21 male, nine female (mean ± SD) age = 22.73 ± 2.69 years, height = 1.73 ± 0.08 m, body

mass = 70.13 ± 12.39 kg, Body Mass Index = 23.41 ± 2.72, time of weekly physical activity = 7.3 ± 2.84 h).

To be eligible, participants had to be between 18 and 30 years of age (to ensure a homogeneous young adult sample and, therefore, minimizing the potential variability associated with age-related neuromuscular adaptations and physical activity levels), not to have experienced lower limb injury or pain within the last year prior to the study, and self-report as physically active, such that they perform at least 90 min of weekly physical activity, assessed using the International Physical Activity Questionnaire (IPAQ). Exclusion criteria included previous participation in any lower limb balance or proprioception exercise program and the presence of known balance impairments such as vertigo, vestibular or central alterations. All participants provided written informed consent and completed a basic information form prior to data collection, which included demographic (age and sex) and anthropometric (height and weight) measures. Participant recruitment was conducted through volunteer calls at the Faculty of Physiotherapy of the University of Valencia by a convenience sampling, since participants were selected based on availability and willingness to participate in the study.

## Ethics committee

This study was approved by the Ethics Committee of University of Valencia (approval number 1271077), and was conducted following the STROBE Statement for cross-sectional studies to ensure transparency and completeness in reporting (*von Elm et al., 2008*).

## Procedures

### Specific instability device

The BB (Blackboard Training, Innenstadt, Germany) (Fig. 1) is a training device designed for monopodal stability work, composed of two wooden boards that, connected by a strap, are rectangular in shape. Its base has a Velcro surface where wooden half-cylinders can be freely placed. Depending on the placement, different types of instability are achieved. This study intended to analyze seven instability conditions (Fore, Sup, Pro, Rear, Inver, Ever, Total) detailed in Fig. 2, and the floor as a stable condition. In Fore, Sup and Pro conditions the BB had the rearfoot fixed and the forefoot was unstable with bar placed in the center, laterally or medially, forcing pronosupination, supination or pronation, respectively. In Rear, Inver and Ever conditions the BB had the forefoot fixed and the rearfoot was unstable with the bar placed in the center, laterally or medially, forcing inversion-eversion of the rearfoot, only inversion or only eversion, respectively. Finally, in the Total configuration both the forefoot and rearfoot were unstable with the bars placed in the center, forcing the pronosupination and inversion-eversion movements of the entire foot.

### Electromyographic signal processing and maximum voluntary isometric contractions

The chosen technique to record muscle activity was surface electromyography (EMG). The analyzed muscles included: soleus (SOL), gastrocnemius medialis (GM), gastrocnemius

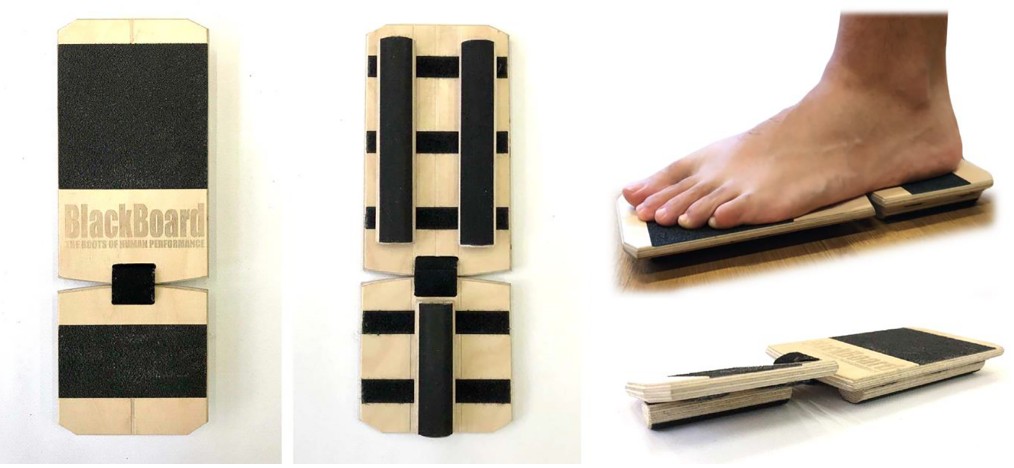

**Figure 1 Selective instability device blackboard.** Photo credit: Mariana Sánchez-Barbadora.

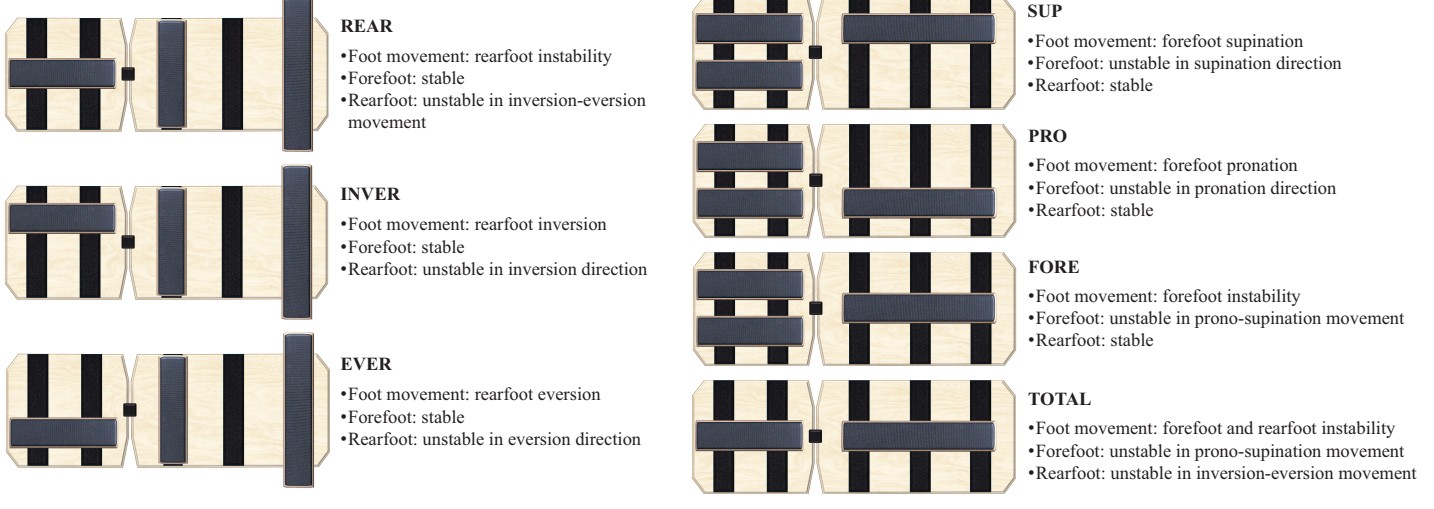

**REAR**
•Foot movement: rearfoot instability
•Forefoot: stable
•Rearfoot: unstable in inversion-eversion movement

**INVER**
•Foot movement: rearfoot inversion
•Forefoot: stable
•Rearfoot: unstable in inversion direction

**EVER**
•Foot movement: rearfoot eversion
•Forefoot: stable
•Rearfoot: unstable in eversion direction

**SUP**
•Foot movement: forefoot supination
•Forefoot: unstable in supination direction
•Rearfoot: stable

**PRO**
•Foot movement: forefoot pronation
•Forefoot: unstable in pronation direction
•Rearfoot: stable

**FORE**
•Foot movement: forefoot instability
•Forefoot: unstable in prono-supination movement
•Rearfoot: stable

**TOTAL**
•Foot movement: forefoot and rearfoot instability
•Forefoot: unstable in prono-supination movement
•Rearfoot: unstable in inversion-eversion movement

**Figure 2 Blackboard configurations for the single-leg stance exercise, for the right foot.**

lateralis (GL), TA, PL, and PB (Fig. 3). After the area was shaved and cleaned with alcohol to minimize skin impedance, adhesive pre-gelled Ag/AgCl EMG electrodes (BlueSensor M; Ambu Ballerup, Denmark; 40 × 34 mm) were placed on the middle region of the muscle belly aligned with the fiber orientation, according to the SENIAM guidelines (Surface ElectroMyoGraphy for the Non-Invasive Assessment of Muscles; www.seniam.org). Verification of electrode placement was done by palpating the muscle while the subject performed a contraction against a resistance provided by the examiner. The lower limb to measure was randomized.

The MuscleLab 4020e (Ergotest Technology AS) EMG device was used to record muscle activity. Raw EMG signals were processed using custom-written scripts in MATLAB (version R2019b; The Mathworks, Natick, MA, USA). First, signals were sampled at

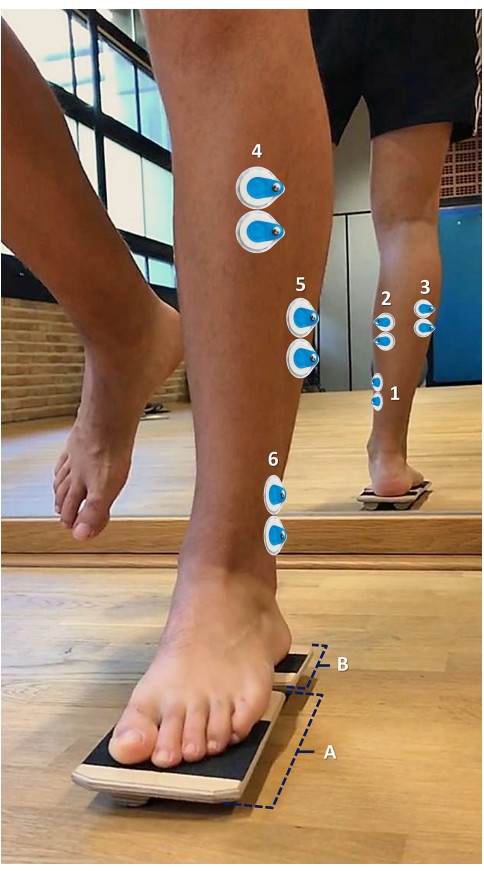

**Figure 3 Electrode placement: (1) Soleus, (2) Gastrocnemius medialis, (3) Gastrocnemius lateralis, (4) Tibialis anterior, (5) Peroneus longus, and (6) Peroneus brevis.** The device (Blackboard) is configured in "Pro" position (*i.e.*, pronation of the forefoot exercise). (A) Forefoot table of Blackboard device. (B) Rearfoot table of Blackboard device.

1,000 Hz. Then, they were band-pass filtered using a fourth-order Butterworth filter with a cutoff frequency of 20 and 500 Hz, rectified, integrated, and converted to root-mean-square signals using a hardware circuit network (frequency response: 450 kHz; averaging constant: 12 ms; total error ±0.5%). To minimize the influence of transient spikes, the EMG signals were visually inspected, and any trials exhibiting abrupt, non-physiological peaks were excluded from analysis.

To allow between-condition comparisons and with normalization purposes, the highest activation of each muscle during a maximum voluntary isometric contraction (MVIC) was recorded. This normalization method has been widely used to compare muscle activation across conditions and individuals (*Besomi et al., 2019*). After a practice trial, each muscle was tested three times for 5 s each, with 2-min intervals between trials. Verbal encouragement was provided during testing (*Harput et al., 2013*).

### Experimental protocol

After a 2-min walking warm-up (*Alfuth & Gomoll, 2018*), subjects were allowed a 20-s familiarization period on the most unstable BB condition (*i.e.*, Total). They stood on their

dominant leg in a single-leg stance on the BB, with the contralateral leg slightly flexed. The single-leg stance condition was selected based on previous research indicating higher activation levels compared to other positions, such as the single-leg squat (*Sánchez-Barbadora et al., 2022*). The dominant leg was chosen following prior studies (*Braun Ferreira et al., 2011*), although its relevance remains debatable (*Petrovic et al., 2022*). The base of the fifth metatarsal was placed on the posterior edge of the front boardof the BB (Fig. 2A), so that the heel rested on the center of the rear board. This placement was chosen because the base of the fifth metatarsal serves as a reference point separating the forefoot from the rearfoot (*Bojsen-Møller, 1979*; *Lee et al., 1999*). Knee and hip were extended, with hands placed on the hips. Subjects were instructed to maintain balance, ensuring that the edges of the balance board did not touch the ground. The sequence of these conditions was randomized.

Three 20-s work periods with 30-s rests were measured for each condition to determine the average (*Alfuth & Gomoll, 2018*). A 2-min rest was given between exercises (*Wahl & Behm, 2008*). Failure was considered, and thus the measurement was repeated if the subject moved the BB, touched the floor with the contralateral foot, or lost their body position. Touching the ground with the edges of the BB was also considered a failure. Subjects were barefoot with eyes open, focusing on a fixed point placed 5 m in front of them. The examiner changed the BB configuration for each condition.

## Statistical analysis

To normalize EMG data, the trial with the highest activation from the MVICs was selected. Of the 20-s trials recorded for each exercise, the maximum signal amplitude within the middle 10 s was extracted and divided by the corresponding MVIC value to obtain the normalized EMG value (nEMG). The average nEMG across the three trials, expressed as a percentage of MVIC, was used for final analysis (*Harput et al., 2013*).

SPSS Statistics 28 (IBM Corporation, Chicago, IL, USA) was used to perform all statistical analyses. Descriptive statistics (means, 95% confidence intervals, standard deviations) were defined. Normality was assessed using the Shapiro-Wilk test. A two-way repeated-measures ANOVA was employed to determine differences between muscles, conditions, and their interaction. Effect size was evaluated with $\eta p^2$ (partial Eta-squared): small ($0.01 < \eta p^2 < 0.06$), medium ($0.06 < \eta p^2 < 0.14$), or large ($\eta p^2 > 0.14$) (*Cohen, 1977*).

Additionally, when a significant main effect or interaction was detected, pairwise comparisons were conducted using *post hoc t*-tests with Bonferroni correction. The number of pairwise comparisons was 28 within conditions and 15 within muscles, resulting in a corrected significance thereshold of $p < 0.00015$ to control for Type I error. However, as SPSS provides adjusted p-values through a mathematically equivalent correction, results were interpreted using the conventional significance threshold of $p < 0.05$. Effect sizes for pairwise comparisons were reported using Cohen's *d*: small ($d = 0.20–0.49$), medium ($d = 0.50– 0.79$), or large ($d \geq 0.80$) (*Cohen, 1977*).

## RESULTS

There was significant main effect for condition ($F(4,89, 141,80) = 38.24$, $p < 0.001$, $\eta p^2 = 0.569$) and muscle ($F(3,56, 103,25) = 10.76$, $p < 0.001$, $\eta p^2 = 0.271$) with condition by muscle interaction ($F(11,70, 339,31) = 8.20$, $p < 0.001$, $\eta p^2 = 0.220$). Multiple differences were found between conditions, independent of the muscle studied, and between muscles, independently of the condition (Tables S1 and S2). In relation to the results of the *post-hoc* tests, multiple differences in nEMG were found between forefoot configurations compared to rearfoot configurations, as well as compared to the Total instability configuration and the floor. Likewise, differences were also found among the different variants of BB within the forefoot (*i.e.*, Fore, Sup, Pro) and within the rearfoot (*i.e.*, Rear, Inver, Ever). Furthermore, multiple differences were found between muscles within each condition, which are detailed below.

### Within-muscle differences between conditions

Group means for each muscle under each condition are presented in Table 1. Between-conditions mean differences and $p$ values, as well as the detailed *post-hoc* results of the significant with effect sizes, are displayed respectively in Table S3 and S4 of the supplementary material. In relation to the most relevant findings: all configuration showed higher activation than the floor and Rear for the SOL, with Total and Fore configurations producing the highest activation ($d = 1.37$ and $d = 1.56$, respectively, when comparing to the floor). In the case of GM, the Total configuration showed higher activation than Rear ($d = 0.64$), but no further significant differences were observed between conditions. For the GL, both Total and Fore conditions showed higher activation compared to most other configurations, with Total producing the highest activation overall ($d = 1.43$ when compared to Rear). Regarding the TA, the Total, Fore, and Sup configurations showed higher activation than the floor and all rearfoot conditions, with Total producing the highest activation ($d = 1.86$ when comparing to the floor). For the PL, the Total, Fore, Pro, and Ever configurations showed higher activation than the floor, Rear, and Sup, with Pro leading to the highest activation ($d = 1.47$ when compared to Rear). Finally, in the case of the PB, the Total, Fore, Pro, and Ever configurations produced higher activation than the floor, Rear, and Sup, with Total showing the highest activation ($d = 2.21$ when comparing to Rear).

Figure S1 of the supplementary material shows an ensemble of the six muscles with their between-conditions differences.

### Within-condition muscle activation patterns

All between-muscles mean differences and $p$ values, as well as the detailed *post-hoc* comparison analyses of the significant with effect sizes, are displayed in Tables S5 and S6, respectively. Figure 4 shows muscle activation pattern for each condition.

In relation to the main relevant findings: in the Ever configuration, both PL and PB showed higher activation than GM ($d = 0.98$ and $0.68$), GL ($d = 0.81$ and $0.66$), and TA ($d = 0.89$ and $0.69$), with PL also showing higher activation than SOL ($d = 0.64$). In the Sup condition, the TA showed higher activation compared to GM, GL, PL, and PB ($d = 1.16$,

**Table 1 Mean nEMG values (%) for each muscle across different configurations of the BB and the floor during the single-leg stance exercise.**

| | Mean (%); 95% CI (SD) | | | | | | | |
|---|---|---|---|---|---|---|---|---|
| | FLOOR | REAR | INVER | EVER | SUP | PRO | FORE | TOTAL |
| **Soleus** | 32.83 [26.83–38.83] (16.08) | 36.78 [31.20–42.36] (14.95) | 43.41 [36.28–50.54] (19.09)* | 49.83 [41.72–57.94] (21.72)† | 49.55 [41.47–57.64] (21.65)† | 47.99 [41.43–54.54] (17.55)† | 51.15 [43.76–58.54] (19.79)† | 56.19 [48.13–64.25] (21.59)‡ |
| **Gastrocnemius medialis** | 35.17 [29.44–40.89] (15.32) | 33.27 [27.46–39.07] (15.55) | 40.90 [34.83–46.97] (16.26) | 44.03 [37.05–51.01] (18.68) | 37.43 [31.47–43.39] (15.96) | 40.61 [33.92–47.3] (17.91) | 41.71 [35.48–47.94] (16.68) | 45.28 [37.45–53.10] (20.95)** |
| **Gastrocnemius lateralis** | 37.39 [30.22–44.56] (19.19) | 36.4 [31.20–41.6] (13.93) | 42.35 [36.26–48.44] (16.3) | 46.29 [40.16–52.42] (16.42)** | 42.47 [35.25–49.68] (19.32) | 53.34 [45.93–60.76] (19.85)† | 52.87 [47.08–58.67] (15.52)‡ § | 56.98 [50.51–63.45] (17.32)‡ § # |
| **Tibialis anterior** | 38.11 [31.67–44.55] (17.25) | 41.04 [34.88–47.19] (16.49) | 50.10 [42.34–57.86] (20.79) | 45.82 [40.56–51.09] (14.1) | 66.29 [58.34–74.24] (21.29)‡ § | 62.15 [54.87–69.43] (19.5)† § | 68.06 [62.27–73.86] (15.52)‡ § | 71.3 [65.33–77.27] (15.99)‡ § |
| **Peroneus longus** | 47.68 [40.80–54.55] (18.41) | 46.98 [41.74–52.22] (14.04) | 52.45 [45.5–59.41] (18.63) | 68.04 [60.48–75.59] (20.24)† | 49.29 [42.27–56.3] (18.79)§ | **75.66 [69.12–82.2] (17.51)‡ #** | 67.64 [62.46–72.81] (13.87)† # | 65.44 [59.34–71.53] (16.32)† # |
| **Peroneus brevis** | 42.37 [37.59–47.15] (12.81) | 37.57 [33.14–42.00] (11.86) | 43.99 [38.20–49.77] (15.49)** | 61.8 [53.85–69.75] (21.29)‡ | 51.71 [44.31–59.11] (19.81)** | 70.31 [64.92–75.69] (14.43)‡ # | 66.9 [60.93–72.87] (15.99)‡ # | **75.95 [70.18–81.72] (15.45)‡ #** |

**Note:**
nEMG, normalized electromyography activity (expressed as a percentage of the maximum isometric voluntary contraction); CI, Confidence Interval; SD, Standard Deviation. The differences found in the *post hoc* test are expressed with symbols: * Differences from FLOOR; ** Differences from REAR configuration; † Differences from FLOOR and REAR configuration; ‡ Differences from FLOOR, REAR, and INVER configurations; § Differences from EVER configuration; # Differences from SUP configuration. In bold, the configurations that produce the highest muscle activation for each muscle.

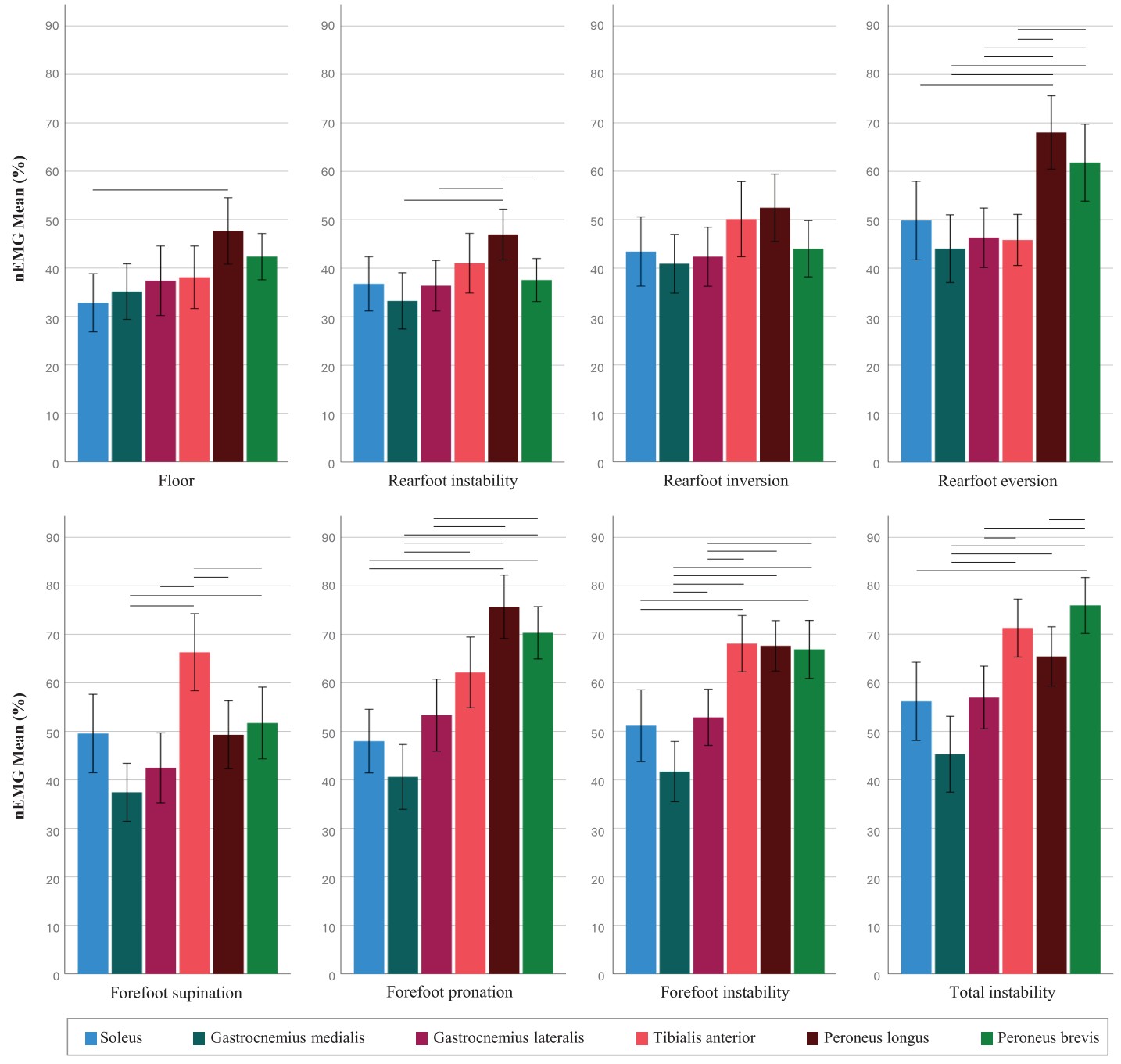

**Figure 4 Muscle activation (% of the maximum voluntary isometric contraction) on the Blackboard in its different configurations.** Horizontal lines represent significant differences.

0.95, 0.73, and 0.59, respectively). For the Pro condition, PL and PB showed higher activation compared to SOL ($d = 0.99$ and $0.95$), GM ($d = 1.49$ and $1.21$), and GL ($d = 0.80$ and $0.71$). In the Fore condition, TA and PB showed higher activation than SOL ($d = 0.67$ and $0.59$), GM ($d = 1.35$ and $1.01$), and GL ($d = 0.66$ and $0.68$), and PL showed also higher

activation than GM and GL ($d$ = 1.10 and 0.77). Finally, in the Total instability condition, TA showed higher activation than GM and GL ($d$ = 1.30 and 0.74), PL was higher than GM ($d$ = 0.87), and PB showed higher activation than SOL, GM, GL, and PL ($d$ = 0.75, 1.43, 1.01, and 0.64, respectively).

## DISCUSSION

This study aimed to compare the muscle activation of six lower-leg muscles during the single-leg stance exercise across different conditions on the floor and various configurations of the BB, and to establish muscle activation patterns for each configuration. Thus, the main findings were, firstly, that all muscles showed multiple differences in activation between conditions, especially the TA, PL, and PB. Secondly, multiple differences were found between muscles in most of the patterns, with the patterns of forefoot supination (Sup) and rearfoot eversion (Ever) configurations being of special interest, as they induced selective activation of the TA and PL respectively compared to the other muscles under study.

These findings align with the study's hypothesis, as the device configurations elicited the activation of the expected muscle, and in most cases, the activation patterns were also the hypothesized (*i.e.*, agonist activation accompanied by inhibition or lack of activation of the antagonist). However, it is worth noting that certain configurations resulted in both the agonist and the antagonist being equally active, with no significant differences between them. This was particularly evident in configurations where the slats were positioned in the middle zone of the device (Fore, Rear, and Total), which specifically induced forefoot pronation-supination, rearfoot inversion-eversion, or a combination of both movements, respectively.

### Muscle activation between conditions

One of the main findings of the present study was that the TA, PL, and PB exhibited the most significant differences between configurations and achieved the highest levels of activation during the single-leg stance exercise (approximately 60% to 80% of MVIC). In addition, the configurations of the BB that seem to be significatively more demanding in terms of muscle activity are Pro, Fore, and Total (forefoot pronation, forefoot pronosupination and pronosupination and inversion-eversion of the entire foot, respectively), with Sup (forefoot supination) and Ever (rearfoot eversion) configurations in certain cases. It was found that, in the case of the TA, the most effective configurations are Sup, Fore, and Total, whilst for activating the peroneus (*i.e.*, PL and PB), Sup configuration is not recommended, and Pro or Ever were chosen instead. Specifically, the Sup configuration promotes tibialis anterior activation by inducing a forefoot supination movement, which requires increased TA engagement to control and stabilize the medial arch. Similarly, the Ever configuration emphasizes peroneus activation by promoting rearfoot eversion, which stimulates the peroneal muscles to counteract instability. This may be related to the type of device being analyzed and plane of motion in which the action takes place. The BB offers a small and rigid base of support, which is partially unstable or firm depending on the configuration. In addition, the configurations under study were

specifically designed to target these stabilizers (*i.e.*, TA, PL, and PB), thereby inducing movement in the frontal plane. *Mayer et al. (2023)*, analyzed kinematics of the leg muscles in a single-leg stance on both sides of the Togu® Jumper® (bladder and flat), suggesting that standing on the bladder side produces greater ankle movement in the sagittal plane, while the flat side does so in the frontal plane (*Mayer et al., 2023*). *Strøm et al. (2016)*, had previously observed a similar effect in terms of inversion-eversion number of directional changes when standing on the Wobble board, which proved to be superior in terms of muscle activation to the BOSU® and Airex. The study proposed that reducing the base of support may be the main cause of this effect (*Strøm et al., 2016*). This aligns with the idea that an unstable but firm small surface can induce movement and elevate activation levels in the mediolateral plane, making therefore the BB an interesting tool to achieve this. Furthermore, the findings of the present study were consistent with the results of *Alfuth & Gomoll (2018)*, who observed differences in the TA and PL with rearfoot stable and forefoot unstable in the frontal plane (equivalent to our Fore configuration) in relation to the floor. However, their study did not analyze variations in forefoot instability from Fore to Sup or Pro (*i.e.*, inducing supination or pronation of the forefoot), nor did they include Total instability configuration or the variations of the rearfoot instability configurations (*i.e.*, Inver or Ever). Therefore, the results presented in this study suggest that other configurations, such as Sup or Total for TA, Pro or Ever for PL, and Pro or Total for PB, could be as effective as Fore in activating each muscle, and may have some additional specific clinical applications, as discussed below.

## Muscular patterns between conditions

There are two relevant findings regarding the specificity of Sup (forefoot supination) and Ever (rearfoot eversion) muscular patterns. First, the choice of Sup configuration seems appropriate when the aim was to activate TA while maintaining activation levels of the peroneus, its antagonists, similar to those produced on the floor (Fig. 3–E). This is significant since, to the best of our knowledge, there are no other unstable devices capable of producing a selective activation of TA without, at the same time, the increased activation of the peroneus muscles. Second, the activation of the peroneus muscles in Ever was higher than that of the other muscles under study, especially in the case of PL, which was significantly higher than TA (Fig. 3–D). This observation is similar to the findings of *Alfuth & Gomoll (2018)* who reported differences in PL activation between a stable forefoot and an unstable rearfoot (similar to our Rear configuration) compared to the floor. In the present study, the differences in Ever configuration were not only existent compared to the floor but also to Rear configuration (inversion-eversion of the rearfoot). It is important to highlight that maintaining balance on the BB configured in Ever induced a demand on the subject similar to that generated in Pro condition (forefoot pronosupination), as the movement primarily requires foot pronation and eversion. However, the peroneus muscles in Ever showed greater activation than TA, which did not occur in Pro configuration. Thus, the findings of this study suggest that an eversion action with a fixed forefoot facilitates the specific activation of PL, minimizing the activity of the other stabilizing muscles. Therefore, Ever configuration acts in a specific way and may be clinically relevant

when selective activation of the peroneus is desired, above that of the other muscles under study. Thus, although the configurations Fore, Pro, and Total may be effective for activating TA and the peroneus muscles, working on the BB in its Sup instability variation or with the rearfoot configured in Ever may offer advantages that should be considered in clinical practice.

## Strengths, limitations and clinical applications

Despite the interesting findings of the present study, several limitations need to be acknowledged. First, the study relied solely on EMG data and analyzed only six superficial lower-leg muscles. Including kinematic data from the BB or analyzing additional muscles, such as the tibialis posterior, would have been of great interest. Second, the sample consisted only of healthy subjects, making it controversial to extrapolate the results to injured populations, as it is known that ankle stabilizers in individuals with chronic ankle instability (CAI) exhibit different activation patterns compared to healthy individuals (*Delahunt et al., 2010*; *Suda & Sacco, 2011*). Third, the analysis was performed only during a single-leg stance exercise. Clinicians should ensure that subjects maintain the position to use these findings as a reliable reference. Lastly, since our criteria did not include any specific requirements regarding participants' vision, it is possible that some may have had difficulty seeing the fixed point during the measurement, which could have altered their balance during the tasks, although no such issues were reported. Thus, future research should aim to replicate these findings with injured samples and include additional muscles while also analyzing movement. Including other exercises in the analysis, such as single-leg squats or forward lunges, could maximize the applicability of the results.

The variety of findings obtained in this study suggests diverse clinical applications. The ability to specifically activate TA could be useful in cases of TA weakness or in foot drop condition (*Stewart, 2008*; *Waseem et al., 2023*). Furthermore, the established relationship between delayed electromyographic activation of the peroneal muscles and ankle sprains (*Linford et al., 2006*; *Hopkins et al., 2009*), as well as the lack of peroneal activation in patients with CAI (*Hopkins et al., 2009*), suggests that incorporating the BB in a prevention training program could increase the activation of these muscles. In addition, a recent study suggests the conservative treatment of hallux valgus through contraction of PL (*Ikuta et al., 2024*), so the BB could be an interesting tool for addressing this condition. Additionally, the selective activation of desired muscles may prevent overuse and fatigue of the others, which is crucial during the early stages of rehabilitation or for patients with significant muscle weakness. Moreover, physiotherapists can utilize the findings from this study to determine how to work with the BB, specifically by selecting the appropriate configuration to achieve the desired activation pattern of the muscles. Overall, the findings of this study demonstrate that the BB's specific design effectively activates foot and ankle stabilizers, making it a valuable tool in rehabilitation programs that require controlled and selective muscle activation.

## CONCLUSIONS

According to the results obtained in the present study, the BB functions as an instability device that modulates muscle activation in six lower-leg muscles analyzed, depending on its configuration. In particular, the forefoot supination configuration (Sup) effectively activates the TA, while the rearfoot eversion configuration (Ever) targets the peroneus muscles, both configurations without significantly altering the activation levels of the other muscles compared to standing on the floor. Additionally, forefoot pronation (Pro), forefoot pronosupination (Fore), and a configuration combining forefoot pronosupination with rearfoot inversion-eversion (Total) were effective in simultaneously achieving high activation levels of the TA, PL, and PB. The choice of the most appropriate configuration should be determined by the clinician based on individual needs.

## ACKNOWLEDGEMENTS

We appreciate the collaboration, time, and effort of all participants who voluntarily took part in this study.

### Funding

The authors received no funding for this work.

### Competing Interests

The authors declare that they have no competing interests.

### Author Contributions

- Mariana Sánchez-Barbadora conceived and designed the experiments, performed the experiments, analyzed the data, prepared figures and/or tables, and approved the final draft.
- Vicente Alepuz-Moner analyzed the data, authored or reviewed drafts of the article, and approved the final draft.
- Noemi Moreno-Segura analyzed the data, prepared figures and/or tables, authored or reviewed drafts of the article, and approved the final draft.
- Rodrigo Martín-San Agustín conceived and designed the experiments, performed the experiments, prepared figures and/or tables, authored or reviewed drafts of the article, and approved the final draft.

### Human Ethics

The following information was supplied relating to ethical approvals (*i.e.*, approving body and any reference numbers):

This study was approved by the Ethics Committee of University of Valencia (approval number 1271077).
## Data Availability

The raw measurements are available in the Supplemental File.

## Supplemental Information

Supplemental information for this article can be found online at http://dx.doi.org/10.7717/peerj.19461#supplemental-information.

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
