# Peer review of "Comparison of lower-leg muscle activation and establishment of muscle activation patterns during single-leg stance under various instability conditions in healthy active subjects: a cross-sectional study"

_PeerJ, doi:10.7717/peerj.19461_

## Round 0.1 · original submission · Major Revisions

The manuscript presents a potentially significant contribution to the field of rehabilitation and muscle activation using a new device, the Blackboard. The reviewers have raised important issues that need to be addressed to meet the journal's standards for publication. The decision for major revisions is based on the need for clarification and enhancement in several areas as outlined by the reviewers.

1. Basic Reporting
The manuscript is generally well-written, but it lacks specific details about the device used in the study, which is crucial for understanding its innovation and application. The article should provide a clear description of the Blackboard, including its design and functionality, as highlighted by Reviewer 2. The introduction and background are adequate but can be improved by explicitly connecting the study's methods and aims with the existing literature, particularly in explaining how the device contributes to the targeted muscle activation in rehabilitation. Several reviewers noted the need for clearer explanations of technical terms and study parameters, suggesting that the manuscript could benefit from a more detailed glossary or expanded descriptions within the text.

2. Experimental Design
The research question is relevant; however, the experimental design lacks specific details that are critical for reproducibility. Reviewer 2's comments about including movement characteristics for each configuration of the device and more detailed procedural descriptions are particularly pertinent. There is a general consensus among reviewers that the methodology section needs significant expansion to ensure that the study can be replicated. This includes more detailed descriptions of EMG setup, electrode placements, and the statistical methods used. The ethical standards and participant recruitment strategy need to be more thoroughly described to conform to the prevailing ethical standards in the field.

3. Validity of the Findings
The statistical analysis is noted to be robust by some reviewers, but there is a need for more detailed reporting of the results, including specific statistical values and a clearer explanation of the findings. The conclusions drawn from the data are appropriate, but they need to be more tightly connected to the empirical evidence presented. Reviewers have suggested that the discussion section could be enhanced by directly linking the study results to the hypotheses and existing literature. Reviewer 5 points out the necessity of including a more detailed discussion on how the findings can be applied in rehabilitation settings, which would strengthen the manuscript’s impact.

Recommendations:
The authors are advised to undertake a thorough revision of the manuscript to address the specific issues raised by the reviewers. This includes expanding the sections on methodology, providing a more detailed description of the Blackboard device, and enhancing the clarity of the results and discussion sections. Figures and tables should be revised to ensure they are clear and fully supportive of the text. Additional literature references should be included to strengthen the argument for the device’s utility in rehabilitation, specifically addressing the gaps noted by the reviewers.

This decision reflects the consensus of the reviewers that while the study is of interest and could potentially contribute valuable insights into muscle activation and rehabilitation, significant revisions are necessary to meet the journal's standards for clarity, reproducibility, and depth of analysis. The authors are encouraged to address all the concerns raised in a revised submission.

Reviewer 1 ·

Basic reporting

no comment

Experimental design

no comment

Validity of the findings

no comment

Reviewer 2 ·

Basic reporting

The current study aimed to investigate muscle activation patterns in six lower-leg muscles during single-leg stances on various configurations of the Blackboard instability device and identify its potential for selective muscle activation to optimize rehabilitation processes.
The study is of interest; however, several points need clarification. Please find the suggestions below:
Abstract
1. The Blackboard is described as a "new configurable instability device," but details about its characteristics or design are vague. A more specific explanation of its functionality would better highlight its novelty.
2. Proprioception is broadly described as a key component of motor control and joint stability. However, its relationship to balance training and muscle activation is not sufficiently explained and could be omitted.
3. Terms such as "Sup" (forefoot supination) and "Ever" (rearfoot eversion) are introduced briefly but lack sufficient explanation for readers unfamiliar with these terms. It is recommended to replace these abbreviations with "forefoot supination" and "rearfoot eversion" for clarity.
Introduction
4. Adding an explanation of how unstable boards facilitate proprioception would add value.
5. The distinction between global instability devices (e.g., BOSU, Wobble Board, Mini Stability Trainer) and newer targeted devices like the Blackboard (BB) could be articulated more succinctly, emphasizing their unique characteristics.
6. While the BB is described as innovative, the justification for its specific configurations or targeted muscle activation patterns being beneficial for rehabilitation is underdeveloped. It would strengthen the argument to connect these features explicitly to clinical benefits, such as faster recovery or improved outcomes. Additionally, speculative claims, such as the BB “focuses directly on the desired muscle without overloading others” and enhances rehabilitation processes, require supporting evidence or references.
7. The sentence in line 87 should be moved to the last paragraph to enhance clarity.
8. The hypotheses lack specificity. For example, the phrase "selectively activate the desired muscle" should specify which muscles are expected to be activated under particular conditions. Similarly, "each configuration... will produce its own pattern of muscle activation" would benefit from clarifying the anticipated differences (e.g., intensity, sequencing, or co-contraction).

Experimental design

Materials and Methods
9. A brief description of the movement characteristics of each configuration (Fore, Sup, Pro, Rear, Inver, Ever, Total) of the BB should be included in the text.
10. Adding the foot movements for each board configuration in Figure 2 would help readers better understand the functions of the BB.
11. The section describing participant preparation with EMG equipment should precede the experimental protocol for better flow.
12. Please verify the correct symbol for partial eta-squared (ηp²).

Validity of the findings

Results
13. More details on ANOVA reporting should be included. For instance:
• "There was a significant main effect for condition (F(df_between, df_within) = 38.24, p < 0.001, ηp² = 0.569) and muscle (F(df_between, df_within) = 10.76, p < 0.001, ηp² = 0.271), with a significant condition by muscle interaction (F(df_between, df_within) = 8.20, p < 0.001, ηp² = 0.220)."
Discussion and Conclusion
14. Using subheadings in the discussion section aligned with the results would improve the structure and help readers follow the narrative more easily.
15. The phrase "He suggested that" in line 247 could be rephrased for clarity and formality. Consider alternatives such as "The study proposed that...," "The authors hypothesized that...," or "It was suggested that...".
16. After terms like "Sup configuration," the corresponding movement (e.g., forefoot supination) should be included in brackets for better understanding.

·

Basic reporting

Clear and unambiguous, professional English used throughout:
In general, the language is appropriate and there are minor revisions to be made (see general comments).

Literature references, sufficient field background/context provided:
The references adequately cover the area of research. In general, the background/context is adequately addressed, but an explanation of why decoupling forefoot and rearfoot movements is necessary for balance training is lacking. I would suggest formulating 1-3 sentences based on the relevant literature.

Professional article structure, figures, tables. Raw data shared:
In general, tables and figures are appropriate and there are minor revisions to be made (see general comments)
Raw data are shared.

Self-contained with relevant results to hypotheses:
Confirmed.

Experimental design

Since this is a cross-sectional study of healthy subjects and no treatment was given, I think the term experimental study is not appropriate. I would suggest using the term quasi-experimental study or just cross-sectional study. For definitions of study types I would refer to the Joanna Briggs Institute (JBI) and their categorisations.

Validity of the findings

In general, yes. However, Bonferroni correction should be specified as described in the statistical methods under General comments.

Additional comments

Abstract
Please do not use abbreviations in the abstract. I would suggest to revise the following sentence into:
Additionally, forefoot supination and rearfoot eversion configurations…

Introduction
• Lines 49-51: I think the term bipedal stance is more common than bipodal. Furthermore, I would suggest to use increasing instead of enhancing, and have been shown to be effective… => Unstable devices are commonly used in single-leg stance or in bipedal stance to improve stability, and have been shown to be effective in increasing lower leg muscle activation during a rehabilitation program.
• Please specify BOSU, Power board, and Mini Stability Trainer, including type of device, manufacturer, city, country, as these devices seem to be from private label manufacturers.
• Line 87: The first objective is not clear to me … to compare the electromyography (EMG) appears not appropriate, because electromyography is the measurement method and not the outcome. The outcome is to compare the muscle activation between the different configurations using electromyography?! Please reformulate or specify your objectives.
• Line 92: repetition of „that“: It was hypothesized that: …., and 2) that…

Materials and methods
• Lines 99/107: I would suggest using body mass in kg here, because body weight is expressed in Newtons, taking into account gravity; please provide Body Mass Index
• Line 100: Please explain why you have chosen your age range.
• Lines 99-103: I would suggest to present weekly physical activity in hours; please define physical activity => according to the definition of e.g. WHO?
• Lines 107-108: Please indicate that your sample was a convenience sample
• Since you conducted a cross-sectional study, did you follow the STROBE-Statement? If no, please consider it.

Statistical analyses
• Since there are a lot of pairwise comparisons, please provide more details on the Bonferroni correction => how many pairwise comparisons and what was the corrected p-level?

Results
• Okay.

Discussion
• Line 224: This study aimed to compare the activity of six lower-leg muscles….
• Line 247: Who is meant by He? Strom? The citation is Strom et al., so it should be They?!
• In general, you switch between simple past and present. For example, in lines 273-275 you used present, but your data analysis is done and you should use simple past here: However, the peroneus muscles in Ever showed greater activation than TA, which did not occur in the Pro configuration => Please read the entire paper on tense.

Reviewer 4 ·

Basic reporting

no comment

Experimental design

1) BB is a medical device?

2) What is the diameter of the electrodes used for sEMG?

3) What is the software used for process EMG signals (e.g. filtering, RMS etc.). It is not clear from the paper whether both the acquisitions and the analyzes are carried out with MuscleLab 4020e

Validity of the findings

1) Broaden the discussion on potential applications of the research in rehabilitation

Reviewer 5 ·

Basic reporting

PeerJ #111412
This manuscript presented lower-leg muscle activation during the use blackboard tools. There are several concerns regarding this manuscript need to be address prior consideration for publication.
General Comments
1. The topic and introduction of this study are interesting and clearly presented.
2. The methodology contains unclear details regarding procedural instructions and data analysis. Please revise these sections according to the specific suggestions below.
3. The summary table is confusing and includes unnecessary information, making it difficult to interpret the findings. If the author intends to publish the entire results, a more detailed discussion is needed.
4. The discussion requires additional detail to better explain the findings.
5. The conclusion effectively summarizes the study's findings.

Experimental design

Specific Comments
Methodology
6. Overall, providing photos of the experimental session, electrode placements, and various tasks performed would enhance reader understanding and significantly improve the study's repeatability and reproducibility.
7. Lines 122–123: Which leg was the participant instructed to use for the single-leg stance in this experiment? What was the rationale for choosing that specific side? Please specify.
8. Lines 123–124: Can you indicate where the posterior edge of the front board is (either in the text or through an illustration)? This clarification would help readers better understand the task procedure. Additionally, why was the fifth metatarsal base selected as the reference point to separate weight-bearing between the front and rear boards?
9. Line 126: According to Sánchez Barbadora et al. (2022), the Blackboard was used to emphasize the activation of the peroneus longus during a single-leg squat. Why did your study use a single-leg stance instead of a single-leg squat? Please justify this design choice.
10. Line 129: If the edge of the balance board touches the ground, does this count as a trial failure? Please clarify.
11. Line 132: Participants were instructed to focus on a fixed point 5 meters ahead. Did the screening criteria account for potential vision issues among participants?
12. Lines 153–155: This study uses the maximum amplitude value to represent muscle activation patterns. How do you confirm that this value accurately reflects overall activation over the 10-second period? Could it be influenced by brief spikes in activity? Additionally, how were outliers in the EMG data identified and handled?
13. Line 170: Consider moving the sample size calculation to the participant section to ensure all relevant details are presented in one location.
14. Figure 1: It would improve clarity if you included an illustration showing a participant performing the task or focused specifically on foot positioning during the experiment. Additional figures may also be helpful.

Validity of the findings

Results
15. Table 1 is incomplete in PDF format. Thus, I could not fully review this portion. However, the table includes multiple symbols and comparisons. Are all these comparisons necessary? For example, are the comparisons between Fore vs. Rear, Ever vs. Inver, or Pro vs. Rear, Inver, and Sup essential? These comparisons are not discussed in the results or discussion sections. If they are unnecessary, please highlight only the primary findings relevant to your study objective.
16. Figure 3: Instead of using symbols to represent differences, it would be clearer to use brackets, making the interpretation easier. Since each graph is already labeled with its respective condition, the additional labels (A, B, C, etc.) are redundant and can be removed.
Discussion
17. Lines 236–238: Please clarify the conditions in greater detail. How do Sup, Fore, and Total conditions emphasize tibialis anterior activation and the peroneus muscle group? Specifically, how do your recommended conditions (Sup for TA activation and Ever for peroneus activation) achieve this effect?

---

## Round 0.2 · accepted · Accept

Two original reviewers have confirmed that their comments have been fully addressed and have recommended the manuscript for publication. I have also reviewed the revised version and find the responses to be thorough and satisfactory. The manuscript is now clear, methodologically sound, and well-organized.

Based on the reviewers' endorsements and my own assessment, I am confident that the current version is ready for publication.

Reviewer 2 ·

Basic reporting

Thank you very much for your thorough revision of the manuscript. The authors have addressed the concerns raised, and the changes have notably improved the clarity and overall presentation of the study. The refinement of the content, terminology, and structure has enhanced the readability and scientific value of the work.

Experimental design

The authors have addressed the previously raised concerns in the Materials and Methods section. The descriptions of the movement characteristics for each Blackboard configuration (Fore, Sup, Pro, Rear, Inver, Ever, Total) are now clearly included in the text, and the corresponding foot movements have been effectively illustrated in Figure 2. Additionally, the order of the experimental protocol has been improved by repositioning the EMG preparation section for better flow. The symbol for partial eta-squared (ηp²) has also been corrected. These revisions enhance both clarity and methodological transparency.

Validity of the findings

The authors have appropriately addressed the suggestion regarding ANOVA reporting.

·

Basic reporting

The authors have addressed all comments appropriately. I have no further comments.

Experimental design

The authors have addressed all comments appropriately. I have no further comments.

Validity of the findings

The authors have addressed all comments appropriately. I have no further comments.

Additional comments

The authors have addressed all comments appropriately. I have no further comments.